# A Survey of Deep Learning Methods for Estimating the Accuracy of Protein Quaternary Structure Models

**DOI:** 10.3390/biom14050574

**Published:** 2024-05-13

**Authors:** Xiao Chen, Jian Liu, Nolan Park, Jianlin Cheng

**Affiliations:** 1Department of Electrical Engineering and Computer Science, University of Missouri, Columbia, MO 65211, USA; 2NextGen Precision Health Institute, University of Missouri, Columbia, MO 65211, USA

**Keywords:** protein quality assessment, estimation of model accuracy, deep learning, protein complex, protein quaternary structure

## Abstract

The quality prediction of quaternary structure models of a protein complex, in the absence of its true structure, is known as the Estimation of Model Accuracy (EMA). EMA is useful for ranking predicted protein complex structures and using them appropriately in biomedical research, such as protein–protein interaction studies, protein design, and drug discovery. With the advent of more accurate protein complex (multimer) prediction tools, such as AlphaFold2-Multimer and ESMFold, the estimation of the accuracy of protein complex structures has attracted increasing attention. Many deep learning methods have been developed to tackle this problem; however, there is a noticeable absence of a comprehensive overview of these methods to facilitate future development. Addressing this gap, we present a review of deep learning EMA methods for protein complex structures developed in the past several years, analyzing their methodologies, data and feature construction. We also provide a prospective summary of some potential new developments for further improving the accuracy of the EMA methods.

## 1. Introduction

Proteins interact to form complexes that carry out important biological functions. Therefore, obtaining the quaternary structure of a protein complex is crucial for elucidating how proteins interact. This information proves useful in addressing various biological research problems that require protein–protein interaction (PPI) details, such as drug discovery [1,2,3] and protein design [4,5].

Typically, high-resolution structures of protein complexes are determined with methods such as X-ray crystallography and cryo-electron microscopy (cryo-EM). However, these low-throughput methods can only solve a small portion of protein complex structures. As a result, computational protein complex (multimer) prediction methods have gained significant attention in the scientific community. Recently, some advanced deep-learning protein multimer predictors, such as AlphaFold-Multimer [6], have substantially accelerated the process of predicting protein complex structures. These tools initially generate a large number of protein-complex decoys. Then, the quality of all the predicted structures (decoys) needs to be assessed (predicted) by a quality evaluation method, which is not aware of the native structure of the protein complex and ultimately selects the decoy with the highest score. Typically, this phase is referred to as the estimation of model accuracy (EMA), protein quality assessment (QA), or model scoring problem. Figure 1 illustrates such a protein complex structure prediction and quality evaluation process. As seen, after an EMA method selects the highest-ranked decoy, that decoy is passed to downstream application tasks.

Numerous methods have been developed to address the challenging problem of estimating the accuracy of protein complex structures [7,8,9,10,11,12,13]. Commonly, EMA methods can be categorized into three types of approaches: physical energy-based [12], statistical potential-based [7,8], and machine learning-based. Physical energy-based methods typically use scoring functions consisting of a weighted linear combination of various physical energetic terms. Due to their lower computational complexity, these models are frequently employed in numerous docking methods [14,15,16]. Statistical potential-based methods convert the distribution of distance-relevant or irrelevant pairwise contacts at the atom or residue level into statistical potentials [8]. Machine learning or deep learning-based methods [11,13,17,18,19,20,21,22,23,24,25,26,27,28,29,30,31] generally use features to represent a protein quaternary structure that are then used to predict the structure’s quality scores. Similar to the protein tertiary structure EMA task, quaternary structure EMA methods can also be classified into multi-model [23,32] and single-model approaches [24,25]. Multi-model methods [22,23] yield a relative score for each predicted protein complex decoy in a model pool, leveraging the similarity between structural models to assess quality. In contrast, single-model methods consider only one decoy and assign an absolute quality score to it, independent of comparisons with other decoys.

In this review, we will concentrate on the deep learning-based protein quaternary structure EMA methods developed in the last five years. The structure of this review is as follows: First, we will introduce the metrics for evaluating the quality of protein complex structures and the performance of EMA methods. Then, we discuss the prevalent methods for representing protein complex structures for EMA and common datasets used to train and test EMA methods. Subsequently, we explore the recent deep learning EMA methods, categorized by their technical approaches. Finally, we conclude the review with a perspective on future trends in the development and enhancement of protein complex structure EMA methods.

## 2. Metrics for Evaluating the Quality of Protein Complex Structures and the Performance of EMA Methods

CASP (Critical Assessment of Techniques for Protein Structure Prediction) and CAPRI (Critical Assessment of PRedicted Interactions) are two worldwide experiments that rigorously test computational methods of predicting protein complex structures and estimate their accuracy. The latest competitions [33] offer widely accepted measures for assessing the overall (global) structural quality, interface quality, and local structural quality of predicted complex structures with respect to their true structures, as well as for evaluating the performance of EMA methods of estimating/predicting the accuracy of predicted complex structures, as discussed below.

### 2.1. Global Structural Quality Evaluation of Protein Complex Structure

CASP uses oligo-GDT-TS [34] and TMscore [35] to evaluate the global structural quality of a predicted complex structure.
(1)$$\mathrm{oligo-GDT-TS}=\frac{GDT_{P8}+GDT_{P4}+GDT_{P2}+GDT_{P1}}{4}$$

**oligo-GDT-TS**: Similar to the GDT-TS (Global Distance Test Total Score) [34] used to evaluate a predicted tertiary structure, oligo-GDT-TS extends the calculation to the global structural similarity score of the predicted complex structures with respect to the true structure. It is defined by Equation (Equation 1), where $GDT_{Pn}$ represents the GDT-TS score at a specific distance threshold (Pn), i.e., the percentage of residues in the model that fall within a certain distance threshold (Pn = 8 Å, 4 Å, 2 Å or 1 Å) from their corresponding residues in the actual structure. This calculation is performed after aligning the predicted and true structures.

**TMscore**: Unlike oligo-GDT-TS, the Template Modeling score (TMscore) rescales residue-wise modeling errors, eliminating the dependency on protein size. TMscore is calculated using Equation (Equation 2), where $L_{target}$ is the length of a target complex structure, $L_{common}$ is the number of the aligned residue pairs, $d_{i}$ is the distance between the Cα atoms of the *i*th pair of residues from the two structures (e.g., a predicted target complex structure and the corresponding true structure), and $d_{0}$ is calculated by Equation (Equation 3) according to the length of the predicted protein complex.
(2)$$\mathrm{TM-score}=\max\left\{\frac{1}{L_{\mathrm{target}}}\sum_{i=1}^{L_{\mathrm{common}}}\frac{1}{1+{\left(\frac{d_{i}}{d_{0}\left(L_{\mathrm{target}}\right)}\right)}^{2}}\right\}$$
(3)$$d_{0}\left(L_{target}\right)=1.24\sqrt[{3}]{L_{target}-15}-1.8$$

### 2.2. Interface Quality Evaluation of Protein Complex Structure

The interface quality measurement is meant to evaluate the similarity between the structure of the residues in the interaction interface of a predicted protein complex structure and the structure of their counterparts in the true structure. DockQ score [36] is a widely used metric to measure interface quality and is calculated based on three complementary measurements: f(nat), L_rms, and i_rms.

f(nat) is the fraction of native contacts in the protein–protein interfaces in the true complex structure preserved in the predicted protein complex structure. Protein–protein interface residues are determined by whether the distance between any two heavy atoms of two residues from two different protein chains is less than 5 Å. This fraction, denoted as f(nat), ranges from 0 to 1, with a higher value indicating a greater preservation of native contacts in the predicted complex structure.

L_rms is the ligand root mean square deviation (RMSD), which is defined as the RMSD of the backbone of a chain of the predicted complex structure, called a ligand (usually the shorter chain) after the structure of another chain called the receptor (usually longer chain) is superimposed with that of the true structure. The larger the L_rms value, the worse the predicted complex structure.

i_rms is the interface root mean square deviation, which is the root mean squared deviation between the residues in the interface region of a predicted complex structure and the true structure. The larger the i_rms value, the worse the predicted complex structure is in comparison with the true structure.

**DockQ**, in the range [0, 1], is the average of f(nat), scaled L_rms, and scaled i_rms, calculated according to Equation (Equation 4). The scaled value of L_rms and i_rms are determined by Equation (Equation 5) with two thresholds ($d_{1}$ and $d_{2}$), respectively. The threshold values for $d_{1}$ (8.5 Å) and $d_{2}$ (1.5 Å) are selected through a grid search to optimize the F1-score for classifying predicted protein complex structures in a dataset into the four quality classes (Incorrect, Acceptable, Medium and High) as commonly used in the CAPRI competition. Additionally, these four quality classes can be further grouped into two main classes: incorrect and correct.
(4)$$DockQ=\frac{f\left(nat\right)+RMS_{scaled}\left(L\_rms,d_{1}\right)+RMS_{scaled}\left(i\_rms,d2\right)}{3}$$
(5)$$RMS_{\mathrm{scaled}}\left(RMS,d_{i}\right)=\frac{1}{1+{\left(\frac{RMS}{d_{i}}\right)}^{2}}$$

In addition to the DockQ score, QS-score [37] is also used to evaluate the interface quality of complex structures. QS-score, expressed by the weighted fraction of shared interface contacts (e.g., residue-residue pairs with Cβ−Cβ distance < 12 Å), is another robust continuous score used to measure the interface quality of a predicted complex structure. It is calculated by Equation (Equation 6), where $d_{A}$ and $d_{B}$ denote the distance between two residues in contact in protein structure A (e.g., predicted structure) and B (e.g., true structure), respectively. The weighting function w(d) is calculated with Equation (Equation 7), which is the probability of a side-chain interaction between two residues given their distance. A higher QS-score indicates a higher similarity between compared interfaces.
(6)$$\mathrm{QS-score}=\frac{\sum_{shared(A,B)}w\left(min\left(d_{A},d_{B}\right)\right)\left(1-\frac{|d_{A}-d_{B}|}{12}\right)}{\sum_{shared(A,B)}w\left(min\left(d_{A},d_{B}\right)\right)+\sum_{non-shared(A)}w\left(d_{A}\right)+\sum_{non-shared(B)}w\left(d_{B}\right)}$$
(7)$$w\left(d\right)=\left\{\begin{array}{cc} 1, & \mathrm{if}\;d<=5.\\ e^{-2{\left(\frac{d-5}{4.28}\right)}^{2}}, & \mathrm{if}\;d>5. \end{array}\right.$$

**CAD-score** [38] is another metric to evaluate interface quality. Let G be the set of residue pairs with nonzero contact areas in a predicted complex or true complex structure. $T_{\left(i,j\right)}$ and $M_{\left(i,j\right)}$ are the contact areas for residue pair (i,j) in the true structure and predicted structure, respectively. The CAD score is calculated by Equation (Equation 8), where the $CAD_{\left(i,j\right)}^{bounded}$ is calculated by Equation (Equation 9) and $CAD_{\left(i,j\right)}$ is calculated by Equation (Equation 10). The higher the CAD score, the better the interface quality is.
(8)$$CAD-score=1-\frac{\sum_{(i,j)\in G}CAD_{\left(i,j\right)}^{bounded}}{\sum_{(i,j)\in G}T_{\left(i,j\right)}}$$
(9)$$CAD_{\left(i,j\right)}^{bounded}=min\left(CAD_{\left(i,j\right)},T_{\left(i,j\right)}\right)$$
(10)$$CAD_{\left(i,j\right)}=\left|T_{\left(i,j\right)}-M_{\left(i,j\right)}\right|$$

### 2.3. Local Structural Quality Evaluation of Protein Complex Structure

The local structural quality of the predicted protein complex structure, i.e., the quality of the predicted position of each residue or each atom in a predicted complex structure, is often evaluated by the Local Distance Difference Test (lDDT) score [39].

**lDDT** quantifies the degree to which a protein structure accurately replicates the environment found in the true structure. For each atom in the true structure, the nearby atoms from different residues are identified by a distance threshold. Then the percentage of the distances between one atom and other atoms in the true structure that are preserved as the predicted structure is calculated. This atom-level lDDT score is the average of the fraction calculated at 0.5 Å, 1 Å, 2 Å, and 4 Å distance thresholds considering all the atoms. In contrast, the residue-level lDDT score is calculated on the Cα atoms only.

All the quality metrics of global structure, interface, and local structure work in the same way for homomers and heteromers.

### 2.4. Metrics of Evaluating Protein Complex Structure EMA

EMA methods are developed to predict the quality score of predicted protein complex structures in the absence of true complex structures. Their performance is evaluated by comparing predicted quality scores with true quality scores, measured by metrics such as those in Section 2.1, Section 2.2 and Section 2.3. Below are some commonly used measures for evaluating the performance of EMA methods.

**Correlation**: the Pearson/Spearman correlation between the true and predicted quality scores of an EMA method.

**MSE and MAE**: mean squared error (MSE) and mean absolute error (MAE) between the real and the predicted quality scores of an EMA method.

**Ranking loss**: the difference between the quality of the actual best structure and the quality of the No. 1 ranked structure, which is selected according to quality scores predicted by the EMA method. The smaller the loss, the better the ranking is.

**Success rate (SSR)**: the percentage of near-native structures among the top *n* ranked structures selected by the EMA method.

**Hit rate (HR)**: the fraction of near-native structures among the top *n* ranked structures, divided by the number of all near-native structures in the structure pool.

## 3. Learning the Representation of Protein Complex Structure

### 3.1. Protein Complex Structure Representation

Deep learning EMA methods take a predicted complex structure as input and predict its quality score. Unlike a vector of numbers that can be directly used as input for deep learning methods, a predicted complex structure consists of a set of atoms. This requires their x, y, and z coordinates to be converted into a representation that can be understood and processed by deep learning methods. Consequently, many different representations for protein complex structures have been developed. iScore [13] utilizes a graph random walk method to construct graph kernel matrices for a given protein complex structure, while CoDES [40] extracts a set of physicochemical and statistical potential features of protein complex structures. Both methods transform protein structures into tableau data that machine learning methods can use.

Dove [17] and TRScore [41] employ a fixed-size 3D grid to encode protein complex structures for model quality classification. PointDE [42] creates a point cloud to represent the protein interaction interface in protein complex structures to assess their quality.

More recently, several methods [20,24,27,28,30,31,43,44,45,46,47,48,49] have used graphs to represent protein complex structures. Compared to other forms, graph structures offer several advantages: (1) they can effectively represent residue–residue or atom–atom interactions, with the capacity to easily assign chemical, physical, biological, and artificial features to the nodes and edges in the graph; (2) graph representations can scale to match protein structures of various complexity and size; and (3) they are particularly suitable for some deep learning algorithms such as graph neural networks while requiring fewer computational resources than 2D and 3D grid representations. Below is a brief description of various graph neural networks used to learn the representation of protein complex structures.

### 3.2. Graph Neural Network

Graph neural networks (GNNs) are specifically designed to process graph-structured data. As a result, when representing protein complex structures as graph structures, GNNs naturally lend themselves to addressing protein complex EMA challenges. Here we provide a concise overview of three common GNN architectures: Graph Convolutional Networks (GCNs) [50], Graph Attention Networks (GATs) [51], and Graph Transformers (GTs) [52]. These architectures form the backbone of numerous protein complex EMA methodologies.

#### 3.2.1. Graph Convolutional Neural Network

A graph convolutional neural network (GCN) is a type of neural network that applies the capabilities of a convolutional neural network to a graph data structure. They provide a particular advantage in making complex neural networks understandable by utilizing the layer-wise propagation Equation (Equation 11) for a specific graph-based neural network model f(X,A):(11)$$H^{\left(\ell +1\right)}=\sigma \left({\tilde{D}}^{-\frac{1}{2}}\tilde{A}{\tilde{D}}^{-\frac{1}{2}}H^{\left(l\right)}W^{\left(\ell \right)}\right)$$

This expression outlines how each layer processes information. $\tilde{A}$ represents the adjacency matrix of the undirected graph *G* with the addition of the identity matrix $I_{N}$ adding self-connections to each node in the graph. ${\tilde{D}}_{ii}$ is the diagonal elements of a degree matrix where *i* is the *i*-th diagonal element of said matrix. ${\tilde{D}}_{ii}$ is computed with the sum of the elements of each row within ${\tilde{A}}_{ij}$, where *i* and *j* represent the element in the *i*-th row and *j*-th column of the matrix. Within the GCN, each layer has its own trainable weight matrix $W^{\left(l\right)}$. The activation function σ(·) is applied to the results of the matrix operations. $H^{\left(l\right)}$ is a matrix that represents node features in the *l*th layer of the GCN where $H^{\left(0\right)}$ equals the initial layer *X* and is updated in each layer using the defined operations.

#### 3.2.2. Graph Attention Neural Network

Graph attention neural networks (GAT) receive an input of nodes h = {$\vec{h_{1}}$, $\vec{h_{2}}$, ⋯, $\vec{h_{N}}$}, $\vec{h_{i}}$$\in R^{F}$ where N is the number of nodes and F is the number of features in each node. To obtain initial transformation features, for each input node, it applies a shared linear transformation, parameterized by the weight matrix W where $W\in R^{F^{\prime }\times F}$ to each node. A self-attention mechanism then computes $e_{ij}$, indicating the importance of node *j*’s features to node *i* using the equation $e_{ij}=a\left(W\vec{h_{i}},W\vec{h_{j}}\right)$. In this formula, *a* represents a weight vector that parameterizes the attention mechanism. Here, the graph structure is incorporated by only computing for nodes $j\in N_{i}$ where $N_{i}$ is some neighborhood of node *i* in the graph. To improve the comparability across nodes, $e_{ij}$ is normalized by the softmax function (Equation (Equation 12)) across all the neighbors of node *i*, resulting in an attention weight $\alpha_{ij}$.
(12)$$\alpha_{ij}=softmax_{j}\left(e_{ij}\right)=\frac{exp(e_{ij})}{\Sigma_{k\in N_{i}}exp\left(e_{ik}\right)}$$

Finally, the layer outputs updated node features h′ = {$\vec{h_{1}^{\prime }}$, $\vec{h_{2}^{\prime }}$, ⋯, $\vec{h_{N}^{\prime }}$}, $\vec{h_{i}^{\prime }}\in R^{F^{\prime }}$ with a potentially different number of features $F^{\prime }$, from the weighted sum of the features of the neighbors of each node.

#### 3.2.3. Graph Transformer Neural Network

The Graph Transformer (GT) neural network adapts the transformer model [53] from graph data processing to update both node and edge features, which is different from GAT above which only updates node features. A graph *G*, characterized by initial node features ${\hat{h}}_{i}^{0}$ and edge features ${\hat{e}}_{ij}^{0}$, is input into the Graph Transformer layer for message passing and feature updating. Initially, the Graph Transformer layer computes the attention scores by utilizing both ${\hat{h}}_{i}^{\ell }$ and ${\hat{e}}_{ij}^{\ell }$. Subsequently, it derives updated intermediate node features ${\hat{h}}_{i}^{\ell +1}$ and edge features ${\hat{e}}_{ij}^{\ell +1}$ based on these attention scores. The outputs are then passed to a feed-forward network, which is proceeded and succeeded by residual connections and normalization layers, leaving the final output the updated node feature $h_{i}^{\ell }$ and edge feature $e_{ij}^{\ell }$.

## 4. Datasets for Training and Test Protein Complex EMA Methods

Training, validation, and testing constitute critical stages in the development of deep learning methods. The quantity and quality of the training and test data are critical for constructing high-performing deep learning methods. Table 1 enumerates some commonly used datasets for training, validating, and testing protein complex structure EMA methods.

The DockGround set [54] provides two docking decoy sets for the EMA benchmark, which are associated with 61 unbound complex targets. Each target has 100 decoy models, including at least one near-native (L_rms<5) decoy generated by a protein docking tool GARMM-X [55,56,57].

The Docking Benchmark set (BM set) is a successful protein–protein scoring benchmark dataset series, with the first DBM set published in 2003 [58] being named DBM1.0. Each later BM set version was built on top of the previous version by adding new targets. The latest version, DBM5.5 [59], contains 230 complex targets, several decoys predicted for them, and the quality scores (labels) of the decoys.

The PPI4DOCK docking set [60] contains 1417 non-redundant docking targets and also provides 54,000 decoys generated by ZDock 3.0.2 [61]. A CARPI set [62] consists of 15 published CAPRI targets, a total of 19,013 decoys generated by 47 different predictor groups. About 10% of decoys are of acceptable or higher quality (based on CAPRI standards). The CASP15 dataset has 38 complex targets, with each target containing around 250 decoys generated by different structure prediction teams. DBM55-AF2 [24], contains 15 targets with a total of 300 decoys generated by AlphaFold-Multimer. DockGroud set, PPI4DOCK, CAPRI set, CASP15, and DBM55-AF2 provide both decoys and native structures for each target. By calculating the quality metric of each decoy with respect to the native (true) structure as labels, decoys can be used for training and testing deep learning models.

For accurately assessing a deep learning method’s performance, the redundancy between the training and test datasets should be removed. Usually, a target is not included in a test set when its sequence has 30% or higher sequence identity than any protein target in the training and validation datasets.

It is worth noting that the size of most datasets above is small and their decoy models were generated by traditional docking methods instead of the state-of-the-art protein complex structure prediction methods such as AlphaFold-Multimer, which may not be sufficient to train large deep learning methods. Therefore, most recent deep learning EMA methods [24,26,27,28,48] have used their own custom datasets for training. For example, DProQA applied AlphaFold2 [63] and AlphaFold2-Multimer [64] to generate complex structures for training. VoroIF-GNN [21] collected 1567 heterodimer structures and used FTDock [65] and FASPR [66] to generate decoys for them for training.

## 5. Deep Learning-Based EMA Methods for Protein Complex Structure

In this section, we review the deep learning-based protein complex structure EMA methods developed within the last 5 years. Table 2, Table 3 and Table 4 list the summary information of each method, including release date, main techniques, predictions, representation (atom/residue) level, single/multi-model method designation, input features, training dataset, testing dataset, and source code’s URL.

TRScore [41] employs a 3D CNN architecture, adapted from a ResNet-inspired VGG [69,70] architecture with structural re-parameterization technique (RepVGG), to predict the likelihood of a near-native model for an input 3D complex structure. When provided with an input 3D structure, a 3D grid with 20×20×20 shape and 2 Å grid spacing is constructed after voxelizing the 40×40×40 cube that is placed on the centroid of the protein–protein interface. Each voxel in the grid is then assigned a 19-dimensional vector feature representing the counts of atoms of different types within the voxel. Evaluated on the BM5 dataset, DockGround unbound decoy set, and CAPRI decoy set, TRScore can obtain a performance comparable to or better than DOVE [17], ZRANK [71], ZRANK2 [72] and IRAD [10] in terms of success rate and hit rate.

DOVE [17] applies two knowledge-based statistical potential values, GOAP [73] and ITScore [9], as the description of the input decoy and also represents the position of carbon, oxygen, nitrogen, and other atoms at the interface. These features were concatenated and reshaped as the fixed cube shape (40 Å × 40 Å × 40 Å) for a 3D CNN model to predict the probability that an input protein complex structure has an acceptable model quality based on the CAPRI standard. The DOVE is trained and validated on the BM4.0 dataset [74] and tested on the DockGround set [54]. Because it uses a fixed-size cube as input, DOVE faces the difficulty of accurately capturing the protein interface of large-size protein complex structures. Also, 3D-CNN is computationally expensive for modeling. To address these issues, the GNN-DOVE [18] is proposed. GNN-DOVE extracts the interface region of the protein complex and then reconstructs a graph with/without inter-molecular interactions to represent it as input. It then predicts a probability that the input protein complex has a CAPRI-acceptable quality. GNN-DOVE was trained on the DockGround and DBM4 set. For a fair comparison with DOVE, DOVE was also retrained on the same dataset as GNN-DOVE. On the CAPRI set [62], GNN-DOVE shows a higher performance in terms of hit rate.

PAUL [44] is an end-to-end protein complex scoring system. PAUL does not use any pre-calculated statistical or physical terms as the input feature for the neural network. PUAL instead uses rotation-equivariant neural networks with three hierarchical structures to represent protein–protein complex atoms’ positions and types. PAUL consists of two models with the same architecture: the first one is a classification model (for ranking purposes) to predict if a decoy has an acceptable quality (i.e., L_rms < 10 Å), and the second one is the regression model (for model quality assessment purpose) to directly predict the LRMSD. PAUL’s training dataset is BM4.0 and evaluated on BM5.0 (excluding the BM4.0 part) and PPI4DOCK set. Because of PAUL’s ranking ability, it is sometimes used to enhance other model scoring methods’ ranking ability. PAULSOAP-PP and PAULZRANK, for example, use PAUL to filter out sub-optimal decoys first and then rank the remaining decoys using SOAP-PP or ZRANK, which perform better than using SOAP-PP and ZRANK alone in both benchmark datasets.

ECGN [12] utilizes two identical GNNs with different parameters to represent intra- and inter-molecular residue–residue contacts in a protein complex structure to predict its binding energy. The energy term is calculated from the interface root mean square deviation (iRMSD) of the complex structure. For both intra- and inter-molecular residue contact graphs, ECGN employs 4 node features and 11 edge features. ECGN was trained on the BM4 dataset and tested on the CAPRI targets and score_set. On the test datasets, ECGN demonstrates a better performance than a random forest-based scoring method on both the ranking task and the quality estimation task and its performance is comparable to iRAD [10].

PIQLE [31] utilizes GAT to predict a single global interface quality score from the interaction interface graph extracted from the input protein complex structure. Each graph is assigned 17 residue-level sequence- and structure-based node features and 27 multimeric geometric-based edge (interaction) features. PIQLE was trained on the DockgGround set and benchmarked on the HAF2 set [24], achieving a better performance than DProQA [24], TRScore, GNN-DOVE, and DOVE.

DGANN [67] employs a deep graph attention neural network to predict the likelihood of a protein complex structure model being a near-native model. The graph for an input structure is constructed by treating residues with encoded physical-chemical properties and Position-Specific Scoring Matrix (PSSM) features as nodes, while edges are established when the minimum distance between any two atoms from two different residues is less than 5 Å. Benchmarked on the BM5.5 dataset, DGANN outperformed ZDOCK, HADDOCK [75], iScore [13] and DOVE-Atom40 in terms of success rate and enrichment factor.

DeepRank_GNN [25] adapts a graph structure to represent a protein dimer structure for interface quality prediction. Similar to DGANN, the construction of the graph involves treating residues with their features (e.g., residue type, residue charge, residue polarity, and PSSM) as nodes, with edges symbolizing the contact between the residues. Different from DGANN, DeepRank_GNN builds two distinct input graphs formed by intra-chain (e.g., a minimum atomic distance less than 8.5 Å) and inter-chain contacts (e.g., a minimum distance between heavy atoms less than 3 Å). These graphs are then inputted into the graph neural network to predict the fraction of native contacts in the input structure. Benchmarked on the CAPRI dataset, DeepRank_GNN performed better than HADDOCK, DeepRank [76], DOVE, GNN-DOVE and iScore in terms of AUC.

DProQA [24] encodes a protein complex structure as a KNN (10 neighbors) graph and feeds it to a gated graph transformer to predict a real-valued quality score of the structure as well as a quality class that it belongs to. The graph’s node and edge features are all directly generated from the input protein complex structure without using any extra information such as multiple sequence alignments (MSAs) and residue residue co-evolutionary features extracted from MSAs. DProQA is trained and tested on the newly developed protein complex datasets in which all structural decoys were generated using AlphaFold2 and AlphaFold-Multimer. In the blind CAPSP15 experiment, DProQA is one of the top performers among all single-model methods in terms of ranking loss [77].

G-RANK [30] is built on top of the geometric vector perceptron–graph neural network (GVP) [78]. GVP uses directed Euclidean vectors to represent the positions of atoms of protein complex structures for downstream machine-learning tasks. G-RANK uses the graph to represent the protein complex structure’s interface, with atom type as node features, and edge direction and length as edge features. Both node and edge features are embedded in a high-dimension feature space. They are then sent to GVP for updating via message passing and predicting an interface quality score $f_{nat}$.

GCPNet-EMA [49] represents a successful implementation of Geometry-Complete Perceptron (GCP) Networks [79] for the protein complex structure EMA task. Given a protein complex structure, GCPNet-EMA constructs a residue-level graph, incorporating initial node features, edge features, and frames derived from the residues’ coordinates. These features are processed through several SE(3)-equivariant GCPConv layers. Subsequently, the model employs its learned fine-tuned representations to predict the lDDT score for each residue (node). When benchmarked on the CASP15 multimer set, GCPNet-EMA demonstrates competitive performance in terms of ranking loss. Interestingly, GCPNet-EMA can be applied to predict lDDT scores for both protein quaternary structures and tertiary structures. Similarly, EnQA based on 3D-equivariant graph neural networks [43] was originally trained on protein tertiary structures to predict per-residue lDDT scores, but can also be applied to predict the lDDT scores of protein quaternary structures by treating them as a single unit.

ComplexQA [48] designs a new graph-based neural network for predicting the local residue quality of protein complex interfaces based on the sequence and three-dimensional structure-derived features. ComplexQA first generated thousands of features and finally selected the top 300 features that had the highest Pearson correlation with the labels. Additionally, to accurately evaluate ComplexQA’s performance, a modified lDDT score—lDDT_C30—is proposed. Compared to the original lDDT score, lDDT_C30 enlarges the default radius from 15 Å to 30 Å when calculating the fraction of local residue pairs. In the experiment, lDDT_C30 shows a higher correlation with the DockQ score than lDDT.

VoroIF-GNN [68] (Voronoi InterFace Graph Neural Network) adapts the Voronoi tessellation of atomic balls of van der Waals radii to establish atomic contacts, which are aggregated to form residue contacts. The inter-residue contacts characterized by their attributes such as contact surface area, contact-solvent border length, sum of inter-contact border lengths, and contact-type descriptors serve as nodes in the constructed interface graph. Meanwhile, the inter-contact borders are designated as edges in this graph, which are fed to a graph neural network to predict the CAD-goodnesses derived from the CAD-score. Evaluated on the CASP15 dataset, VoroIF-GNN was the best single-model method in terms of ICS, IPS and lDDT-oligo [77].

PointDE [42] first extracts the interaction interface from a protein complex structure and then converts it to an interface point cloud. To sample a fixed number of points from the initial interface points vector, PointDE proposes a new Nearest Pair Sampling approach to sample a fixed number of atom pairs on the interface. As a result, each final point cloud contains 500 closest atomic pairs. Each point has 3D coordinates and a 26-dimensional feature vector. PointeDE takes PointMLP [80] as the backbone and uses the Inter-molecular Group Mechanism (IGM) to replace the k-Nearest Neighbor (kNN) algorithm to aggregate geometric features during the grouping process. PointDE predicts the probability that a protein complex is native-like or not.

DeepUMQA3 [27,45] extracts overall complex features, intra-chain (intra-monomer) features, and inter-chain (inter-monomer) features for a given complex structure. The overall features are generated by considering the whole complex structure as a single component, which contains ultrafast shape recognition (USR), voxelization expression, inter-residue distances and orientations, and amino acid properties. The sequence embedding generated by a protein language model [81], secondary structure, and Rosetta energy terms [82] of each monomer are considered as the intra-chain features. The inter-chain features are composed of an attention map between the monomer sequence and inter-monomer USR. All these features are first combined and then updated by a triangular multiplication update layer, an axial attention layer, and a feed-forward layer to generate higher-level features. These higher-level features are fed to a residual neural network to predict the local residue quality score and interface residue accuracy. DeepUMQA3 ranked first in the accuracy estimation for protein complex interface residues in CASP15 [77].

GraphGPSM [46] represents a protein complex as a residue-level graph and uses the same features of DeepUMQA3 [45] with the additional coordinates of the $C_{\alpha }$ atoms of the input complex structure to embed the graph. The graph is then updated by the Equivalent-GNN (EGNN) to predict the global TM score of the complex structure.

GraphCPLMQA [47] combines the protein language model-generated sequence- and structural-embedding features, triangular location, reside-level contact order, and physicochemical properties of protein complex structure as the initial features, which are used by a graph neural network-based encoding module to generate high-level features. The high-level features are used by a CNN-based decoding module to predict the residual-level local quality score.

## 6. Performance of Some EMA Methods in CASP15

Although there is no benchmark for evaluating all the EMA methods reviewed in this article on a common dataset in the field, four of them, i.e., VoroIF-GNN (CASP15 group name: VoroIF), DeepUMQA3 (group name: GuijunLab-RocketX), GraphGPSM (group name: GuijunLab-Threader), and DProQA (group name: MULTICOM-egnn), participated in the EMA experiment of CASP15 in 2022, providing an evaluation of their relative performance. In the CASP15 experiment, three metrics were applied to evaluate the EMA predictors: SCORE, QSCORE, and Local Score [77]. SCORE integrates TMscore and oligo-GDT-TS to assess the global topological accuracy of predicted structures. QSCORE combines DockQ and QS-score to evaluate the accuracy of the interface, while the Local Score metric uses the lDDT and CAD scores to measure local interface accuracy.

In the CASP15 experiment, among the single-model EMA methods, GraphGPSM achieved the highest ranking based on the SCORE metric. VoroIF-GNN excelled in estimating interface accuracy in terms of QSCORE, and DeepUMQA3 demonstrated the best performance in estimating local interface accuracy.

## 7. Future Work

Although important contributions have been made towards the estimation of protein complex structure accuracy as discussed above, few methods can consistently estimate the accuracy of protein complex structure models better than the built-in quality scores assigned to them by protein complex structure prediction methods (e.g., the confidence score of AlphaFold-Multimer) [23]. One reason for this sub-optimal performance is the lack of large labeled protein complex structure datasets to train and test deep learning EMA methods.

Public datasets available for training protein complex structures, including the CAPRI set, Docking Benchmark dataset, and Dockground set, typically contain a limited number of structural models (decoys) for a small number of protein complex targets. These datasets cover only a small portion of the protein structure and sequence space. Furthermore, many datasets contain decoys not generated by the latest high-accuracy protein structure predictors, such as AlphaFold-Multimer, potentially causing misalignment between the deep learning model’s training and inference stages. Also, most protein complex targets of these datasets are dimers, with only a small portion of them dedicated to multimers (i.e., more than two chains). This imbalanced distribution could degrade the performance of deep learning models in estimating the accuracy of multimer models. To address these disadvantages, some recent EMA methods have started constructing custom datasets generated by state-of-the-art protein complex structure predictors. These feature a more diverse model distribution in terms of length and number of chains. However, very large public high-quality datasets for the protein complex structure EMA task are still lacking. In addition to the need to create large datasets of protein complex structures, leveraging large datasets for protein tertiary structures that are significantly more abundant in predicted structure databases such as AlphaFold DB [83] and ESM Metagenomic Atlas [81] to train deep learning methods for predicting the quality of protein complex structures can be a viable approach. As demonstrated by GCPNet-EMA and EnQA, deep learning methods trained on tertiary structures can be applied to quaternary structures. One can first train an EMA method on very large datasets of protein tertiary structures and then fine-tune it on a dataset of protein quaternary structures.

In addition to overcoming the major bottleneck of lacking large datasets in the field, another direction is to explore more sophisticated deep learning methods to represent and process protein complex structures. Recently, most EMA methods have represented protein complex structures as graphs for training deep learning models or for feature extraction. Compared to other representations, graph structures provide a more flexible way to encode protein complexes and easily assign features. We expect that more sophisticated graph neural network-based EMA methods will be developed in the future.

## Figures and Tables

**Figure 1 biomolecules-14-00574-f001:**
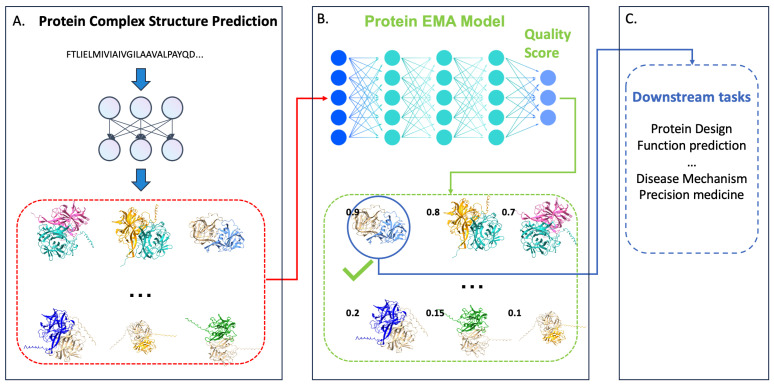
Protein complex structure prediction and evaluation pipeline. (**A**) Protein complex structure predictor generates a pool of structural models (decoys) from the sequence of a protein complex. (**B**) Protein EMA method evaluates (predicts) the quality of predicted decoys. (**C**) The high-quality decoy selected is used for downstream application tasks.

**Table 1 biomolecules-14-00574-t001:** The six common protein complex benchmark datasets. All the datasets except the Docking Benchmark contain both true and predicted structures. The Docking Benchmark dataset includes only true (native) structures. *: None.

Data Sources	Number of Targets/Structures	Source
DockGround	61/6100	http://dockground.bioinformatics.ku.edu/
Docking Benchmark	230/*	https://zlab.umassmed.edu/benchmark/
PPI4DOCK	1417/54,000	http://biodev.cea.fr/interevol/ppi4dock/
CARPI set	15/19,013	http://cb.iri.univ-lille1.fr/Users/lensink/Score_set/
CASP15	38/9930	https://predictioncenter.org/download_area/CASP15/predictions/oligo/
DBM55-AF2	15/450	https://zenodo.org/record/6569837

**Table 2 biomolecules-14-00574-t002:** Summary of EMA methods for protein complex structures (Part 1). *: Paper accepted time.

Name	Year *	Main Techniques	Prediction	Representation Level	Single-/ Multi-Model
PAUL [44]	2020	Equivariant-GCN	iRMSD	Atom	Single
DOVE [17]	2020	3D-CNN	The probability of an input decoy has an acceptable quality or not	Atom	Single
EGCN [12]	2020	GCN	iRMSD	Residue	Single
GNN_DOVE [18]	2021	GAT	The probability of an input decoy has an acceptable quality or not	Atom	Single
DGANN [67]	2021	GAT	The probability of an input decoy is near-native or not	Residue	Single
Trscore [41]	2022	3D-CNN	The probability of an input decoy is near-native or not	Atom	Single
DeepRank_GNN [25]	2022	GNN	f-nat (fraction of native contacts)	Residue	Single
VoroIF-GNN [68]	2023	GAT	CAD score	Atom	Single
DeepUMQA3 [27,45]	2023	2D-CNN	lDDT	Residue	Single
DProQA [24]	2023	GT	DockQ	Residue	Single
G-RANK [30]	2023	GVP	f-nat (fraction of native contacts)	Atom	Single
PIQLE [31]	2023	GAT	Interface score, Fold score, Residue score	Residue	Single
GraphGPSM [46]	2023	EGNN	TM-Score	Residue	Single
GraphCPLMQA [47]	2023	GT + EGNN + 2DCNN	lDDT	Residue	Single
PointDE [42]	2023	Point cloud network	The probability of an input decoy is near-native or not	Atom	Single
ComplexQA [48]	2023	GCN	Interface residue score	Residue	Single
GCPNet-EMA [49]	2024	GCP	lDDT	Residue	Single

**Table 3 biomolecules-14-00574-t003:** Summary of EMA methods for protein complex structures (Part 2).

Name	Features
PAUL	Atomic positions and types
DOVE	Contact potentials, GOAP, ITScore
EGCN	Node features: side-chain pseudo atom’s charge, non-bonded radii, and distance-to-Ca, solvent accessible surface area (SASA). Edge features: atom distance features
GNN_DOVE	Node features: atom physicochemical proprieties of atoms. Edge features: covalent bonds, atom distance.
DGANN	Node features: physical-chemical properties, PSSM, information content
Trscore	Atoms’ physicochemical features
DeepRank_GNN	Node features: residue type, residue charge, residue polarity, buried surface area, PSSM; conservation score, information content, residue depth, residue half-sphere exposure. Edge feature: residue distance
VoroIF-GNN	Node features: contact surface areas, contact-solvent border length, sum of inter-contact border lengths; contact type-dependent descriptors. Edge feature: inter-contact border length
DeepUMQA3	Ultrafast Shape Recognition (USR), residue voxelization, inter-residue distance and orientations, amino acid properties; level of intra-monomer: sequence embedding, secondary structure, energy terms; inter-monomer level: attention map of the inter-monomer paired sequence, inter-monomer USR
DProQA	Node features: residue type, secondary structure type, relatively accessible surface area, torsion angles, node positional encoding. Edge features: Three types of distance, edge positional encoding, contact indicator, permutation-invariant chain encoding
G-RANK	Node features: atom types; edge features: edge direction, edge length
PIQLE	Node features: residue encoding, relative residue positioning, secondary structure, SASA, torsion angles, number of effective sequences (⁠Neff). Edge features: multimeric interaction distance, multimeric interaction orientation
GraphGPSM	USR, residue voxelization, inter-residue distance and orientations, amino acid properties; level of intra-monomer: sequence embedding, secondary structure, energy terms; inter-monomer level: attention map of the inter-monomer, paired sequence, inter-monomer USR, Ca coordinates
GraphCPLMQA	MSA embedding, sequence embedding, structure embedding, triangular location and residue-level contact order, relative position encoding, dihedral and planar angles, voxelization and distance map, Meiler, Blosum62 and DSSP
PointDE	Atomic type, residue types and coordinates, chain identity
ComplexQA	Sequence features, three-dimensional structural and chemical features
GCPNet-EMA	Node features: residue type, positional encoding, virtual dihedral and bond Angles over the $C_{\alpha }$ trace, residue backbone dihedral angles; Residue-wise ESM embeddings, residue-wise AlphaFold 2 plDDT, residue-sequential forward and backward vectors; Edge features: Euclidean distance between connected $C_{\alpha }$ atoms, directional vector between connected $C_{\alpha }$ atoms

**Table 4 biomolecules-14-00574-t004:** Summary of EMA methods for protein complex structures (Part 3).

Name	Training Data	Testing Data	Source
PAUL	DBM4	DBM5, PPI4DOCK	NA
DOVE	DBM4	DockGround	https://kiharalab.org/dove/
EGCN	DBM4	CAPRI	https://github.com/Shen-Lab/EGCN
GNN_DOVE	Dockground, DBM4	CAPRI	https://github.com/kiharalab/GNN_DOVE
DGANN	DBM4	DBM5.5	https://github.com/coffee19850519/PPDocking/tree/master
Trscore	DBM4	DBM5	https://github.com/BioinformaticsCSU/TRScore
DeepRank_GNN	DBM5	CAPRI	https://github.com/DeepRank/Deeprank-GNN
VoroIF-GNN	Custom set	Custom set	https://www.voronota.com/expansion_js/
DeepUMQA3	Custom set	Custom set	http://zhanglab-bioinf.com/DeepUMQA/
DProQA	Dockground, DBM5.5, Custom Dataset	Custom Dataset	https://github.com/jianlin-cheng/DProQA/tree/main
G-RANK	DBM5	CAPRI	https://github.com/ha01994/grank
PIQLE	Dockground v2	Dockground v1	https://github.com/Bhattacharya-Lab/PIQLE
GraphGPSM	Custom set	CASP15	http://zhanglab-bioinf.com/GraphGPSM/
GraphCPLMQA	Custom set	CASP15	http://zhanglab-bioinf.com/GraphCPLMQA/
PointDE	DOCKGROUND	CAPRI, Custom Dataset	https://github.com/AI-ProteinGroup/PointDE
ComplexQA	DockGround, DBM5, Custom Dataset	Custom set	https://github.com/Cao-Labs/ComplexQA/tree/main
CGPNet-EMA	Custom set	CASP15, Custom set	https://github.com/BioinfoMachineLearning/GCPNet-EMA

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
