# Peer review of "A Survey of Deep Learning Methods for Estimating the Accuracy of Protein Quaternary Structure Models"

_biomolecules, 2024, doi:10.3390/biom14050574_

Round 1
Reviewer 1 Report
Comments and Suggestions for Authors
This is a nicely written review on an important topic in the field of structural biology, namely the prediction of quaternary structure. To my knowledge, the list of methods addressed in the work is comprehensive.
Out of curiosity, I wonder whether there's any meaningful difference in the evaluation and assessment of quaternary structure for homo- vs heteromeric complexes. Perhaps the authors could introduce a short comment on this aspect.
Author Response
Thank you for the great question. All evaluation metrics we discussed in the manuscript work same for homo and heteromeric complexes. We added a sentence regarding this issues into the main manuscript as follows:
“All the quality metrics of global structure, interface, and local structure work in the same way for homomers and heteromers”. The change is highlighted in red in the main manuscript.
Reviewer 2 Report
Comments and Suggestions for Authors
I carefully read the review article submitted by Xiao Chen et al. The authors describe recent progress in estimating protein quaternary structure model accuracy and the impact of deep learning methods. To predict specific PPIs, homology modeling as well as ab initio docking methods are long-standing and have been extensively studied. Recently deep learning neural network architecture-based methods have been developed that can model protein complexes in addition to individual proteins. However, as the author mentioned, the quality of the predicted quaternary structures needs to be assessed which remains a challenge without any prior knowledge or native structures. Therefore, it is imperative to identify the gaps in EMA across various approaches such as physical energy-based, statistical potential-based, or machine learning-based. The presented review is clear, concise, well-written, and provides a comprehensive overview of EMA methods to those working in the field of protein structures and their interactions.
A minor correction: line#249 DockGroup ïƒ DockGround. Line#252 include latest reference to GRAMM method (DOI: 10.1007/978-1-0716-3441-7_5)
Author Response
Thank you for the great suggestions. We fixed the typos by changing DockGroup to DockGround. The change is highlighted in red.
Moreover, we added the recommended reference for GRAMM. The new reference is #57 in Reference section.
Reviewer 3 Report
Comments and Suggestions for Authors
The manuscript by Chen et al. proposes a review of the approaches to score protein complex models based on deep learning. The subject is timely and interesting. The manuscript is well written.
Major:
However, I have a major concern about its content. As presently, the manuscript is mostly a descriptive list of approaches that have been considered to develop new model estimators. While describing the principles of these approaches, including the data-sets used to derive them, there is no information about their relative performance. How will a reader get information about which directions are of interest and which are not ? Can the first sentence of the section 6 be considered as correct in the light of the manuscript content? Where is the impact analysis of the approaches claimed in the abstract? I strongly believe that a section comparing the performance of the methods must be inserted.
Minor:
The description of the approaches is sometime elliptic. E.g. equation 1: oligo-GDT-TS: what do the GDTp8,4,2,1 correspond to? (applicable to many other descriptions in the manuscript).
Author Response
Major concern:
However, I have a major concern about its content. As presently, the manuscript is mostly a descriptive list of approaches that have been considered to develop new model estimators. While describing the principles of these approaches, including the data-sets used to derive them, there is no information about their relative performance. How will a reader get information about which directions are of interest and which are not ? I strongly believe that a section comparing the performance of the methods must be inserted.
Response:
Thank you for the great suggestion. We added a Section 6 (Performance of some EMA methods in CASP15) to address this issue as follows:
“Although there is no benchmark of evaluating all the EMA methods reviewed in this article on a common dataset in the field, four of them, i.e., VoroIF-GNN (CASP15 group name: VoroIF), DeepUMQA3 (group name: GuijunLab-RocketX), GraphGPSM (group name: GuijunLab-Threader), and DProQA (group name: MULTICOM-egnn), participated in the EMA experiment of CASP15 in 2022, providing an evaluation of their relative performance. In the CASP15 experiment, three metrics were applied to evaluating the EMA predictors: SCORE, QSCORE, and Local Score [77]. SCORE integrates TMscore and oligo-GDT-TS to assess the global topological accuracy of predicted structures. QSCORE combines DockQ and QS-score to evaluate the accuracy of the interface, while the Local Score metric uses the lDDT and CAD scores to measure local interface accuracy.
In the CASP15 experiment, among the single-model EMA methods, GraphGPSM achieved the highest ranking based on the SCORE metric. VoroIF-GNN excelled in estimating interface accuracy in terms of QSCORE, and DeepUMQA3 demonstrated the best performance in estimating local interface accuracy.”
We hope the description above provides readers with some information about the relative performance of some top performing methods in a blind experiment, even though a full rigorous evaluation of all the methods on a well control benchmark is out of the scope of this review article. The newly added section is highlighted in red in the main manuscript.
Concern:
Can the first sentence of the section 6 be considered as correct in the light of the manuscript content?
Response:
Thank you for pointing this issue. We have revised the first sentence as “Although important contributions have been made towards the estimation of protein complex structure accuracy as discussed above” to mitigate the concern.
Concern:
Where is the impact analysis of the approaches claimed in the abstract?
Response:
Thank you for comment. We revised the sentence about the impact analysis as follows:
Addressing this gap, we present a review of deep learning EMA methods for protein complex structures developed in the past several years, analyzing their methodologies, data, features construction.
Minor concern:
The description of the approaches is sometime elliptic. E.g. equation 1: oligo-GDT-TS: what do the GDTp8,4,2,1 correspond to? (applicable to many other descriptions in the manuscript).
Response:
Thank you for identifying the problem. We added the following description about the metric:
“It is defined by the equation 1, where GDTPn represents the GDT-TS score at a specific distance threshold (Pn), i.e., the percentage of residues in the model that fall within a certain distance threshold (Pn = 8, 4, 2 or 1 angstrom) from their corresponding residues in the actual structure. This calculation is performed after aligning the predicted and true structures”.